# Impact of Single-Walled Carbon Nanotube Functionalization on Ion and Water Molecule Transport at the Nanoscale

**DOI:** 10.3390/nano14010117

**Published:** 2024-01-03

**Authors:** Alia Mejri, Nicolas Arroyo, Guillaume Herlem, John Palmeri, Manoel Manghi, François Henn, Fabien Picaud

**Affiliations:** 1Unité de Recherche SINERGIES, UFR Sciences et Techniques, Centre Hospitalier, 16 Route de Gray, 25030 Besançon, Francenicolas.arroyo@univ-fcomte.fr (N.A.); guillaume.herlem@univ-fcomte.fr (G.H.); 2Laboratoire Charles Coulomb (L2C, UMR CNRS 5221), Université Montpellier, Place Eugène Bataillon, 34090 Montpellier, France; john.palmeri@umontpellier.fr (J.P.); francois.henn@umontpellier.fr (F.H.); 3Laboratoire de Physique Théorique (LPT, UMR CNRS 5152), Université Toulouse III—Paul Sabatier, 31062 Toulouse, France; manghi@irsamc.ups-tlse.fr

**Keywords:** simulations, nanofluidics, carbon nanotube

## Abstract

Nanofluidics has a very promising future owing to its numerous applications in many domains. It remains, however, very difficult to understand the basic physico-chemical principles that control the behavior of solvents confined in nanometric channels. Here, water and ion transport in carbon nanotubes is investigated using classical force field molecular dynamics simulations. By combining one single walled carbon nanotube (uniformly charged or not) with two perforated graphene sheets, we mimic single nanopore devices similar to experimental ones. The graphitic edges delimit two reservoirs of water and ions in the simulation cell from which a voltage is imposed through the application of an external electric field. By analyzing the evolution of the electrolyte conductivity, the role of the carbon nanotube geometric parameters (radius and chirality) and of the functionalization of the carbon nanotube entrances with OH or COO^−^ groups is investigated for different concentrations of group functions.

## 1. Introduction

Since the end of the 20th century, the development of nanofluidic systems has required extensive research, centered on the understanding of fluids and ion transport within nanoscale channels. Over the last two decades, numerous unexpected features of fluids and ions in these channels have been revealed, diverging significantly from expectations based on macro or microfluidic models [1,2]. These unexpected observations include enhanced water flow [3], the unexpected low dielectric permittivity of water [4], and ionic Coulomb blockade behavior [5]. This research field owes its success not only to advancements in nanofabrication technologies, enabling the creation of well-characterized artificial channels at the nanoscale, but also to progress in the techniques and instruments for studying ion transport [6].

These investigations have unveiled new possibilities in nanofluidics, ranging from membrane science for seawater desalination [7,8] to osmotic energy conversion [9]; unique chemical sensing devices [10,11]; FET (field effect transistors) fabrication [12,13]; and construction of biomimetic nanodevices [14]. The rapid evolution of nanofluidics is further propelled by the strong increase in computational power and the discovery of novel nanomaterials, offering potential platforms for both the experimental characterization and the theoretical simulation of fluids confined in nanocavities. Various “one”-dimensional (such as carbon or boron nitride nanotubes) and “two”-dimensional (carbon or boron nitride nanoslits) materials have been employed in fabricating nanofluidic devices that impose nanometric confinement for fluids. Nanopores can be constructed through porous 2D materials or artificial holes created by focused ion beam (FIB) drilling, while nanochannels can be either assembled from nanosheets into laminates due to the relatively weak interlayer van der Waals force [15], or built according to a physico-chemical protocol.

The use of single-wall carbon nanotubes (SWCNTs) can lead to smart devices with an ideal geometry, since the corresponding nanochannels have diameters close to the size of the solvated ions, as well as high aspect ratios and clean surfaces. The surface conductivity of the channel should be able to be independently and easily modulated with a high intrinsic ionic charge. From this, a peculiar coupling emerges exclusively at the interface of water and carbon materials, leading to unexpected phenomena like ultrafast water flow [16] and new phase transitions [17]. These phenomena not only challenge established theories, but also present promising opportunities for practical applications of carbon materials. Despite challenges in the preparation of 2D carbon materials, such as monolayer exfoliation and controllable perforation, SWCNTs, with their smooth hydrophobic inner walls and nanometer-scale diameter, stand out as an ideal platform for investigating nanofluidic behavior [18].

These hollow cylindrical carbon nanochannels can have easily controllable and highly scalable surface potentials. They also have other major assets coming from their tubular structure, such as a significant capacity for transporting fluids and ions in their internal space. On the other hand, the carbon walls of these nanotubes can either be conductive (metallic) or semi-conductive depending on their crystalline structure. The surface charge of these nanotubes can be easily influenced by an external electrical field directly influencing their intrinsic ion conductivity. Another important feature of these structures is that they exhibit a rapid transport of water and ions thanks to the very smooth internal walls, which favors the movement of species with minimal friction [19,20,21,22,23].

It has also been established that SWCNTs can be functionalized at their surfaces and at their ends, making them highly selective for permeating species, which can be “forced” to enter the channel. Moreover, it is now possible to modulate the surface charge of these nanotubes, as well as the nature and the volume of the chemical functions to be grafted at their ends. It then becomes possible to estimate the impact of these grafted chemical functions on what enters and exits these nanotubes. For instance, for increased cation selectivity, COO^−^ moieties can be used at both entrances of SWCNTs [23].

SWCNTs with diameters ranging from 0.7 to 2 nm are known to offer an optimal electrostatic interaction between the channel surface and the translocating ions, allowing one to overcome the electrostatic screening caused by polar solvents, such as water. Furthermore, SWCNTs with sub-1 nm diameter are expected to present new physical behavior specific to the subcontinuum regime, such as ionic Coulomb blockades [24,25,26], or specific to the unusual organization of water molecules in such geometrically constrained conditions.

In a previous work, we designed a nanofluidic system combining SWCNTs with two graphene sheets. An external potential difference was applied to the system in order to evaluate the transport of solvated ions moving inside the carbon nanotube. Measurements of the ionic current established through the internal area of the tube were undertaken. The key geometrical parameters of the carbon structure (radius and length) were also varied, revealing a peculiar dependence of the ionic conductivity on each studied parameter [27].

In the present study, the effect of the chemical functionalization of the dangling atoms at the entrance of the tube is studied, using different SWCNT chiralities and diameters. The size and the charge of the chemical functions are varied in order to determine the role of steric blockage on the conductivity properties of this nanofluidic device [28]. Since the impact of the aperture on the ionic conduction may depend both on the steric and electrostatic features of the grafted function, many different functional groups could be studied. When carbon nanotubes are open, using for instance oxygen plasma or strong acidic treatment, one expects the carbon atoms located at the tube extremities to be oxidized. For this first investigation, we have selected two functions with potentially interesting features: the carboxylate group (–COO^−^), which is stable at pH = 7, i.e., the electrolyte pH in this study, and the alcohol functions (–OH). This allows us also to analyze the influence of non-charged chemical functions, compared to highly charged ones, on the ionic conductance behavior. To do this, an external electric field is applied during the simulations along the nanopore axis, acting as a supplementary force on each charged atom of the system and mimicking the potential voltage.

## 2. Materials and Methods

In this study the simulated nanopore is a single-walled carbon nanotube defined by Hamada (*n*,*m*) indices with a fixed-length *L* equal to 10 nm. The SWCNT is combined with two graphene sheets located at its ends connecting two reservoirs of water and Na^+^ and Cl^−^ ions at a concentration equal to 1 M. Each rectangular reservoir has the following dimensions: (4.9, 4.9, 4.0) nm^3^. We considered three armchair SWCNTs (denoted (10,10), (11,11) and (13,13)) with increasing diameters of 1.34, 1.49 and 1.76 nm, respectively. The corresponding zig-zag SWCNTs (with Hamada indices equal to (19,0) and (23,0)) are also analyzed in the same conditions to study the influence of the chirality on the ionic transport through the SWCNT. One simulated system is shown in Figure 1.

It has been reported in the literature that entrance properties should play a significant role in ion transport inside both CNTs and graphene oxide nanopores of small diameter [29,30]. The interest in studying the impact of grafted chemical functions, e.g., –OH or –COO^−^, at the inlets of SWCNTs arises because they are almost inevitably created when the tube is opened by the oxygen plasma process [31]. It is known that the exact chemical nature and density of functional groups cannot be experimentally controlled during the opening process. On the contrary, in the simulations, the ionic currents can be calculated by changing the number (or type) of chemical groups (–OH/–COO^−^), at the CNT ends. The number and steric properties of these groups can thus be evaluated as functional groups controlling the entrance of the molecules into the CNT.

In neutral and non-functionalized CNTs, the impact of access resistance is more pronounced the shorter the tube length, and it can be amplified by the presence of the graphene sheets on the tube extremities. In functionalized CNTs, this resistance can be enhanced by the presence of chemical functions, owing to both steric and electrostatic effects. We have previously shown, by means of potential of mean force (PMF) calculations, that the presence of chemical functions usually changes the potential distribution and the energy barrier at the entrance [27], quantities that are considered to be fundamental factors governing ion transport at the nanoscale. We observe that for the same tube helicity, the conductivity increases as the diameter of the tube increases, because it can be assumed that the effect of the functional groups is reduced the larger the diameter of the tube. It was also established in our previous work that this increase in conductivity agrees with the theoretical conductivity in neutral pores, which is dependent on the pore radius for a fixed length. The calculated conductivity is higher when the SWCNTs are charged since the counter ions are able to fill the nanotubes more easily and therefore to contribute to the conductivity of the system. The presence of functional groups at the SWCNT entrance, however, also leads to an increase in the barrier that needs to be overcome by the ions to enter the tube. Here, we focus our study on the role of the chemical functions added on the nanochannel extremities to saturate the dangling bonds of the system created during the nanotube opening step. Several cases are analyzed where not only the size of the chemical functions, but also their concentration, is varied in order to understand the influence of such modifications on the nanotube ionic conductivity. Several systems have therefore been created, differing by the total number of functions added to the SWCNT ends, by the SWCNT diameter, by the presence or not of electrical charges on the SWCNT carbon atoms, and finally by the helicity.

The purpose of this molecular dynamics (MD) study is then to assess the influence of edge chemical functional groups on the entry/exit capacity of water and ions within the carbon structure. The dangling carbon atoms are saturated with –OH or –COO^−^ molecules. The Lennard–Jones potential, bond, and angle (without dihedrals) contributions, describing C–C and C–O forces, are also taken into account in the simulation parameters (Table 1). These contributions were taken from the Charmm36 force field and can be used to express the interaction energy of the system as:(1)$$U_{total}=\sum_{bonds}K_{b}{\left(r-r_{0}\right)}^{2}+\sum_{angles}K_{a}{\left(\theta -\theta_{0}\right)}^{2}+\;\sum_{atoms}\sum_{i<j}\left\{\sqrt{\varepsilon_{i}\varepsilon_{j}}\left[{\left(\frac{\sigma_{i}+\sigma_{j}}{2r_{ij}}\right)}^{12}-{\left(\frac{\sigma_{i}+\sigma_{j}}{2r_{ij}}\right)}^{6}\right]+\left(\frac{q_{i}q_{j}}{r_{ij}}\right)\right\}$$
where $r_{ij}$ is the distance between atoms *i* and *j*, $q_{i}$ is their charge, and other parameter definitions and values are given in Table 1.

Sodium and chloride ions are modeled by the Lennard–Jones and Coulombic potentials, and the parameter values are taken from [32]. A uniform partial charge is introduced on all carbon atoms with a surface charge density for the whole CNT set to σ=−0.038 C/m^2^, which is within the range of the expected values from theoretical observations [1]. To investigate the role of the functional groups leading to the saturation of the dangling bonds at the end of the SWCNT, 0, 5, and 7 (or 8) hydroxyl (or carboxylate) groups were attached at each end of the SWCNT. These numbers of chemical groups could be experimentally realized based on various factors, such as the nature of the chemical sites, electrical repulsion, and edge deformation [2].

The electro-neutrality of the system is ensured in all cases by adding counter-ions to the system to balance the excess of charge in the system. Note that, during the different simulations, all the carbon atoms of the SWCNT are kept fixed except the chemical groups on the extremities.

The TIP3P water model [33] was used to simulate the water behavior in order to maintain reasonable computational times with respect to all the tested parameters and other more sophisticated water models (which increase by a factor of 2 the computational times, at the least). The chosen water model exhibits partial atomic charges centered on the hydrogen and oxygen atoms (Figure 2a). Its rigid geometry is consistent with water in the liquid phase and reasonably predicts the density and enthalpy of vaporization of water under ambient conditions [34,35]. The Charmm36 force fields were originally developed to study protein folding with the TIP3P water model [36]. Nevertheless, these force fields are known to overestimate the diffusion constant of water [27,37]. The values of the force field parameters for the TIP3P model are given in Figure 2b.

During this study, all the molecular dynamics (MD) simulations were performed using the NAMD 2.12 code [38]. MD simulations were conducted in the NPT ensemble with constant pressure during the thermalization and production phases (40 nS). The NVT ensemble was implemented when an external electric field was applied, maintaining a constant volume throughout the simulation to avoid a variation in the resulting voltage during the simulations.

The temperature was set to 300 K using Langevin dynamics. The particle mesh Ewald summation method (PME) [39] was used to calculate the full-system periodic electrostatic contributions with an integration time step equal to 1 fs. Standard Charmm36 force field parameters were applied to the system [40], and water/SWCNT interactions were modeled according to the classical combination rules for the Lennard Jones potential model [41]. The external electric field was modeled by applying electric forces to each ion in the direction of the nanopore axis. Voltage drops from 0.05 to 1 V were applied to study the ionic current in the NVT ensemble. For each simulation, 320 steps of minimization were first launched before running 5 nS of equilibration of the full system. The production runs were then performed for 40 nS at each voltage. Each simulation was repeated 3 times with different initial conditions to avoid statistical bias and to estimate the standard errors.

## 3. Results

For each system investigated in this paper and characterized by its diameter, helicity (armchair and zigzag) and kind of functionalization, we changed only one geometrical parameter of the SWCNT at a time. For each case, the current-voltage characteristic was determined by simulating five different voltages (each repeated three times for statistics) applied to the system, allowing us to deduce five ionic currents. From the linear fits of the *I = f*(*V*) curves shown in Figure 3, the conductance G of the system was determined from the slope of the curve.

Figure 3 groups together some of the different *I = f*(*V*) plots extracted from the simulations for armchair SWCNTs with different radii and functionalized ends and with a surface charge density of σ=−0.038 C/m^2^. The common feature of all these *I-V* plots is their linearity in accordance with our previous study of uncharged SWCNTs [27]. Note that the SWCNT surface charge increases the ionic conductance from 0.5 nS when no charge is present on the SWCNT wall to 1.7 nS when carbon atoms are charged for the (13,13) SWCNT (as one example). The same observation also holds for the other SWCNT radii [27].

We can see in Figure 3a that all the SWCNTs presenting dangling bonds at their ends are those with the highest values of conductance compared to SWCNT where functionalization has been added. For instance, the conductance reaches 1.7 nS for the (13,13) SWCNT, while it drops down to 1.2 nS (resp. 0.8 nS) and 0.8 nS (resp. 0.6 nS) for low (resp. high) –OH or –COO^−^ functionalization. The same trends can be noticed for the (11,11) and the (10,10) SWCNTs. The SWCNT radius R thus plays an important role because SWCNTs with the highest radii present also the highest electric current values, with or without functionalization. Moreover, we observe that the degree of functionalization and the kind of functionalization affect directly the values of the current. For the same SWCNT radius, the conductance tends to decrease from 1.2 nS to 0.8 nS when the degree of functionalization increases, depending on the degree of –OH functions (Figure 3b top and down) or the type of functionalization (–OH or –COO^−^) at the same degree (Figure 3b,c on top). The size and charge of the functionalized groups affect the conductance differently depending on the SWCNT radius. Indeed, we observe for smaller SWCNT radius an increase in the current, while it is slightly reduced for higher SWCNT radii.

Increasing the number of functionalized groups tends to decrease the conductance values, whatever the radius values. For instance, the conductivity for the (10,10) SWCNT decreases from 0.16 to 0.04 nS (resp. from 0.40 nS to 0.16 nS) when the degree of –OH (resp. –COO^−^) functionalization increases. The simulated conductances for different charged SWCNTs are tabulated in Table 2.

To analyze quantitatively the measured conductances, it is important to notice that since the SWCNTs are rather short, the access resistance $G_{a}^{-1}$ (or the access conductance $G_{a}$) must be taken into account following
(2)$$G_{exp}^{-1}=G_{a}^{-1}+G_{p}^{-1},$$
where $G_{exp}$ is the experimental (or simulated) conductance, and $G_{p}$ is the pore conductance given by [27]
(3)$$G_{p}=\frac{\pi {\left(R-R_{vdW}\right)}^{2}}{L}\kappa_{p}.$$

The nanopore has a radius *R* and length *L*, and $R_{vdW}$ is the van der Waals radius associated with the hydrophobicity of the CNT. As shown in Ref. [27] for simulated neutral SWCNTs, a reasonable value is $R_{vdW}$ = 0.3 nm. The pore conductivity $\kappa_{p}$ is taken to be approximated by [29]
(4)$$\kappa_{p}=e^{2}(\mu_{+}+\mu_{-})c_{s}\sqrt{1+{\left[\frac{\sigma }{\;e(R-R_{vdW})c_{s}\;}\right]}^{2}}-\frac{\sigma }{e(R-R_{vdW})c_{s}\;}(\mu_{+}-\mu_{-})$$
where *e* is the electron charge, $\mu_{\pm }$ the ionic mobilities, and $c_{s}$ the ionic concentration in the reservoirs. We have neglected the electro-osmotic contribution, which is usually much lower than the electrophoretic one. Using Equation (4) and the values $\mu_{+}=5.6\times 10^{11}$ s/kg and $\mu_{-}=7.1\times 10^{11}$ s/kg for NaCl [42], one obtains the access conductance values shown in Table 3.

One clearly sees that for a given SWCNT, the access resistance increases when the entrances are functionalized. Neutral –OH groups lead for both the (10,10) and the (13,13) SWCNTs to a higher access resistance (or lower access conductance) than the negatively charged –COO^−^ groups. Indeed, since the CNT is negatively charged, the conductance is mainly due to the transport of sodium ions in the nanopore. Therefore, the negative charge of –COO^−^ groups increases the access conductance of positive sodium ions. Furthermore, when the SWCNT radius increases, the access conductance increases, as expected.

Interestingly, for the sodium ions, the increase is linear following (see Figure 4)
(5)$$G_{a}\left(R\right)=2\kappa_{a}\left(R-R_{0}\right),$$
where $R_{0}≅0.65$ nm, meaning that the access resistance would be infinite with the –OH groups if the CNT radius were equal to this value, whatever the density. The conductivities $\kappa_{a}$ have been obtained by performing linear fits, leading to $\kappa_{a,low}=5.2$ S/m and $\kappa_{a,high}=2.6$ S/m, values that are much lower than the bulk conductivities $\kappa_{b}=e^{2}(\mu_{+}+\mu_{-})c_{s}=19.6$ S/m. This result confirms that for such a small system, the usual (purely geometrical) Hall formula [43] for access conductance, $G_{a}(R)=2\kappa_{b}R$, obtained by applying the concepts of standard fluid mechanics is not valid. Indeed, both the concept of a fluid particle and the nanometric size of the reservoirs forbid the use of a simple continuous approach. Note that for the case without functionalization, we obtain $\kappa_{a}=11.6$ S/m and $R_{0}≅0.63$ nm, but the three points are less aligned, leading to larger error bars.

Performing the same analysis for the case of the –COO^−^ groups is more questionable, first because we only have two points for each degree of functionalization, and second because the negative charges lead *a priori* to a more complex dependence of the access conductance on *R*. By performing a linear fit, Figure 4 (right) shows that the values of $R_{0}$ are smaller, 0.33 nm and 0.6 nm, for low and high degrees of functionalization. This might be because these groups are more flexible than the –OH ones. In contrast to the neutral –OH groups, the access conductivity $\kappa_{a}$ increases for the high degree of functionalization (1.4 S/m), compared to the low one (0.9 S/m). This is certainly related to the larger negative charge in this case.

To complete our study, the role of SWCNT chirality was also investigated for two zig-zag geometries presenting the same radius as the corresponding armchair one. Figure 5 represents the theoretical *I = f*(*V*) curves for the (23,0) SWCNT in different configurations. We tested in this case only the uncharged, charged, slightly and strongly –OH functionalized SWCNTs. As depicted in Figure 5, the SWCNT presenting dangling bonds and a charged wall is the most ion conducting, while the uncharged SWCNT did not present a high ionic current. When the dangling bonds are saturated with low or high –OH concentration, the observed currents are intermediate.

To be as exhaustive as possible, the (19,0) SWCNT was also tested. Results are summarized in Table 4. The radius of this latter SWCNT is smaller than for the (23,0) one, which reveals a global decrease in the SWCNT conductivities in all situations, as observed for the armchair geometries.

To explain the differences in conductivity when the charges are modified on the carbon wall, we have plotted the water and ion densities (Figure 6) along the pore axis as a function of the radial position of the entities studied. These distributions are calculated per unit volume considering the total volume occupied by the species along a cylinder portion of radius *r*, sampling size *dr* (here equal to 0.1 Å) and length *L* (equal to 10 nm). To compare the two situations (uncharged and charged pore walls), we normalized each distribution by its maximum obtained in the uncharged case. As a consequence, in the uncharged case, the distribution will never be larger than 1, and the distributions could be higher or lower than 1 depending on whether or not the charged tube has the ability to be filled with counterions. As shown in Figure 6, we can first observe that water filling does not depend on the charges on the SWCNT. The two distributions are superimposed upon each other. On the contrary, the addition of surface charge improves the filling of the tube with sodium ions due to the presence of negative charges on the tube wall, which forces the system to neutralize the charge of the inner SWCNT. Note also that only a few chloride ions enter the charged SWCNT compared to the uncharged one. Indeed, for this concentration and nanopore radius, the negative surface charge is completely neutralized by sodium ions, forbidding the entry of chloride ions into the nanopore. This is in agreement with the theoretical approach of Ref. [32], which shows that when both $\sigma *=\sigma \frac{\pi dl_{B}}{2e}$ and $\overset{˜}{c}=c\frac{\pi l_{B}d^{2}}{4}$ are smaller than 1, one enters the good co-ion exclusion (GCE) regime where only counter-ions, here sodium ions, enter the pore. For the (10,10) CNT, we have σ*=0.34 and $\overset{˜}{c}=0.56$ (at 1 M), and chloride ions are therefore excluded from the pore, whereas for the (13,13) CNT we get σ*=0.47 and $\overset{˜}{c}=1.07$, the onset of the so-called bulk regime, where a few chloride co-ions can enter the CNT.

Moreover, the modification of the conductivity behavior between large and small tube radii and –OH to –COO^−^ chemical functions is due to the charge of the added groups. Indeed, the analysis of the ion filling of the CNT functionalized with –COO^−^ shows that for the (10,10) SWCNT, the sodium ions are present in higher number (13 Na^+^ ions filled the SWCNT on average) compared to the (10,10) SWCNT functionalized with –OH groups, where only 9 Na^+^ are present in the nanotube. This leads in this case to an increase in the conductivity when going from –OH to –COO^−^ systems. For the (13,13) SWCNT, the quantity of Na^+^ inside the nanotube did not vary (11 Na^+^ on average in the two different situations), and the conductivities decrease when the SWCNT is functionalized by –COO^−^ compared to –OH, due to steric effects. Figure 7a,b,c,d shows, respectively, the autocorrelation function of the number of ions $S(z_{i},\tau =∆t)$, the average occupation time $\left\langle \Delta t\left(z_{i}\right)\right\rangle$, the average sodium number $\left\langle N\left(z_{i}\right)\right\rangle$ at each nanotube axial position $z_{i}$ of the simulation (outside or inside the SWCNT), and $S(z_{i},\tau )$ for $z_{i}$ corresponding to the entrance of the nanochannel, defined as
(6)$$S(z_{i},\tau )=\frac{1}{T}\sum_{k=1}^{T}N\left(k+\tau ,z_{i}\right)N\left(k,z_{i}\right)-{\left\langle N\left(z_{i}\right)\right\rangle }^{2}$$
(7)$$\left\langle \Delta t\left(z_{i}\right)\right\rangle =\frac{1}{T}\sum_{k=1}^{T}\frac{N\left(k,z_{i}\right)}{\left\langle N\left(z_{i}\right)\right\rangle }\Delta t\left(z_{i}\right)$$
where $N\left(k,z_{i}\right)$ is the number of sodium ions at position *z_i_* at time *k* (the unit time in these equations corresponds to a time step equal to 10 ps) and $\Delta t\left(z_{i}\right)$ is the time spent by a sodium ion located at $z_{i}$. These functions have been computed for two cases, (10,10) and (13,13) SWCNTs, functionalized by −5 –COO^−^ chemical groups at each end. As we can see on the tube extremities, the autocorrelation function of the sodium atoms and the average number of sodium ions present a peak at the entrance of the functionalized SWCNT (more pronounced in the (10,10) case compared to the (13,13) one). This is because there is more room available at the pore entrance, thus increasing the average and the fluctuations of the number of sodium ions at this specific highly charged position. This function almost vanishes progressively inside the SWCNTs due to the diffusion of the sodium atom in a regular unidimensional chain inside the nanotube.

The occupation time presents a regular behavior outside and inside the two SWCNTs. However, we can notice that the sodium ions move faster inside the (13,13) SWCNT compared to the (10,10) one for which the occupation time is higher. The presence of higher peaks on the autocorrelation function and multiple peaks in the occupation time for the (10,10) case could indicate that the sodium ions are more present at the extremities of the SWCNT and stay longer in these positions during the simulations compared to the (13,13) case. In other terms, sodium atoms can release rapidly from these ends when the nanotube diameter is higher. Figure 7d shows that the relaxation time at the pore entrance for the (10,10) CNT is around 100 to 200 ps, larger than the first one, estimated between 50 to 100 ps, for the (13,13) CNT (even if the data are quite noisy). This could explain the modification of the conductivity behavior in the latter cases. A second larger relaxation time on the order of 400 ps also appears, which might be associated with the presence of sodium ions that stay longer at the pore mouth to screen the charge of the functionalization groups, leaving room for other faster sodium ions, which can then pass through the center of the pore mouth. This effect is not seen in the thinner (10,10) CNT because the pore mouth is blocked.

The influence of carbon nanotube helicity on conductivity cannot be clearly established at this level of investigation. Indeed, our simulations show similar values of conductivity for the (13,13) and (23,0) tubes, as well as for the (11,11) and (19,0) tubes. We can, however, notice a modification of about 2 to 20% (depending on the number of functions) between the (13,13) and (23,0) SWCNTs in the same conditions of functionalization, while it reaches 10 to 50% for the (11,11) and (19,0) SWCNT ones. The most important differences are reached for the highest degree of functionalization. While it appears significant, the low statistics of the simulations for these cases suggests that these results should be verified experimentally because ion entrance events are very low in these cases. (Since the metallic or semi-conducting properties of the SWCNTs are not taken into account in our MD simulations, only the purely geometrical aspects of these chiral SWCNT have been simulated.) Sam et al. [44] performed, however, NEMD and EMD in order to estimate the friction coefficient, slip length and surface potential energy at the water–CNT interface, and their results provided insight into the interaction between water molecules and the inner surface of the cage. These results show that the difference in conductivity values between armchair and zigzag tubes may originate from the variations in the potential energy landscape felt by the water molecules toward the inner surface of the tube. In the armchair (13,13), tube peaks and wells occur alternately along the z and θ direction. For the zig-zag (23,0) tube, the peaks and wells align to form continuous ridges and valleys along the θ direction with an alternating arrangement along the z direction, thereby creating a water flow obstruction in many areas of the tube. These properties seem to be at the origin of the different conductivity values.

## 4. Conclusions

We have conducted an extensive numerical and theoretical study involving the impact of different chemical CNT functionalizations on the ionic conductance of a confined electrolyte. Our results show that the ionic conductance is strongly modified when the dangling bonds are saturated with charged –COO^−^ groups, while it is less reduced when uncharged –OH groups are grafted to the extremities of the CNT nanochannel. The influence of CNT chirality and radius has also been analyzed. Although we have found essentially no influence of the chirality, the radius has been found to be a fundamental parameter that influences the ionic conductance, a result corroborated by our theoretical investigations. These simulation results can be verified experimentally and could be an important working database for future studies aimed at understanding why nanofluidic experiments lead to differences in electrolyte conductivity, depending on the experimental conditions, especially those related to the opening of the tubes.

## Figures and Tables

**Figure 1 nanomaterials-14-00117-f001:**
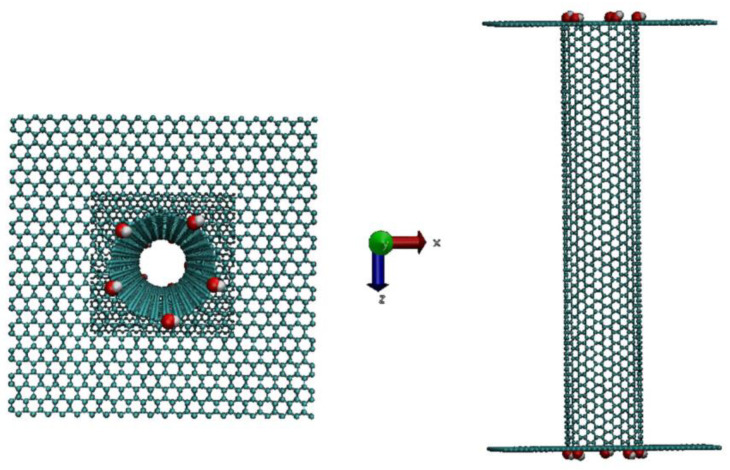
System (10,10) SWCNT functionalized by –OH functions. Top view (**left**) and side view (**right**) before optimization.

**Figure 2 nanomaterials-14-00117-f002:**
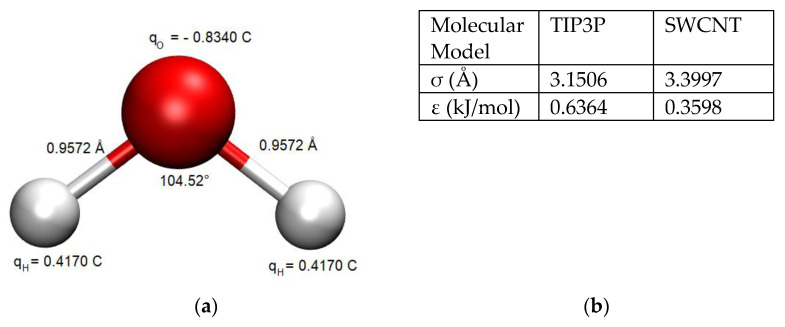
(**a**) Representation of TIP3P. Oxygen and Hydrogen atoms are shown in red and in white, respectively. (**b**) Van der Waals parameters for water model and SWCNT atoms.

**Figure 3 nanomaterials-14-00117-f003:**
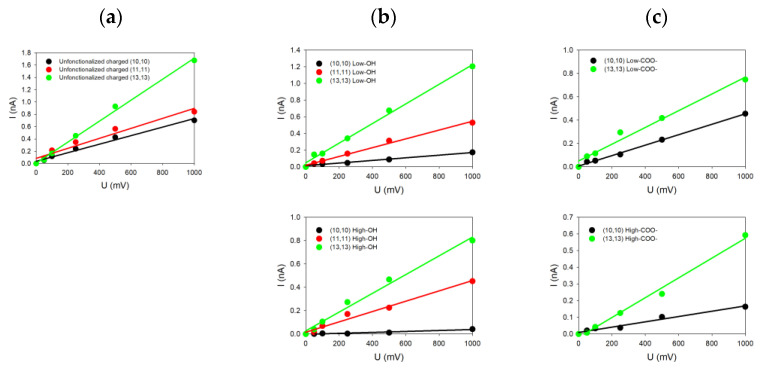
(**a**) Theoretical curves *I = f*(*V*) for different charged SWCNT with ends (**a**) unfunctionalized; (**b**) functionalized with –OH (top: low functionalization rate; bottom: high functionalization rate); (**c**) functionalized with –COO^−^ (top: low functionalization rate; bottom: high functionalization rate).

**Figure 4 nanomaterials-14-00117-f004:**
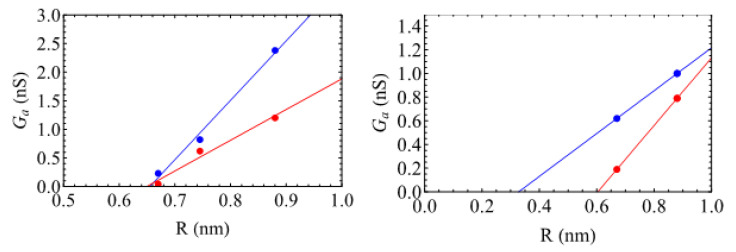
Access conductance $G_{a}$ (in nS) as a function of the SWCNT radius *R* (in nm) with ends functionalized with –OH (**left**) and –COO^−^ (**right**) at low functionalization degree (blue) and high functionalization one (red). The lines correspond to linear fits (values given in the text).

**Figure 5 nanomaterials-14-00117-f005:**
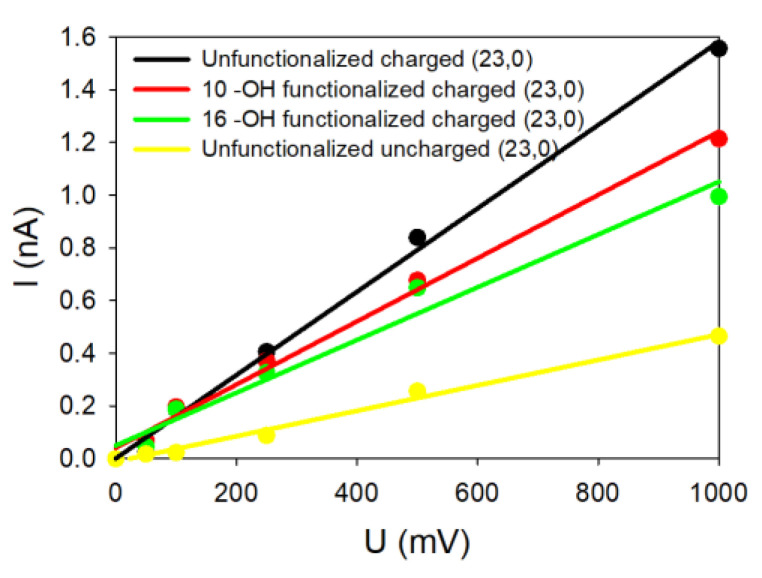
Theoretical curves *I = f*(*V*) for different (23,0) SWCNT.

**Figure 6 nanomaterials-14-00117-f006:**
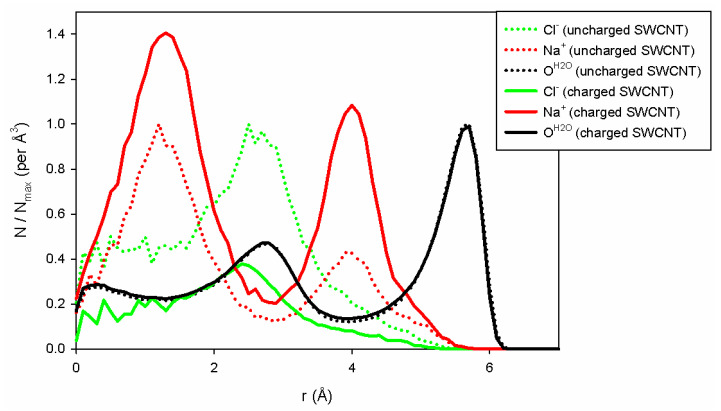
Water and ion distribution per elementary volume inside (13,13) CNT (Cl^−^ ions are in green lines; Na^+^ in red and water in black lines). Results have been obtained for uncharged (dotted) and charged carbon atoms (full lines), respectively. The ion distributions were normalized according to the maximum value of each distribution in the uncharged case, in order to compare the uncharged and charged SWCNT correctly.

**Figure 7 nanomaterials-14-00117-f007:**
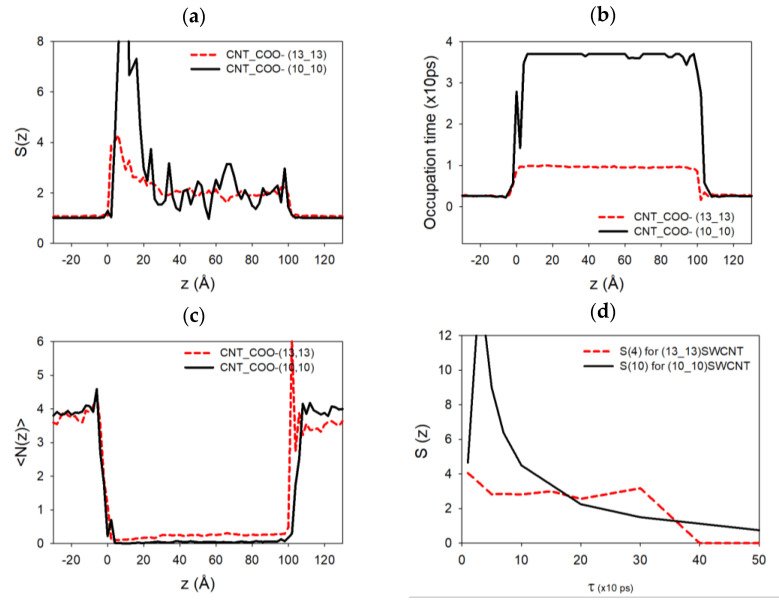
(**a**) Autocorrelation function of the number of sodium ions as a function of the position along the tube axis. (**b**) Average occupation time (over the total simulation time) of each axial tube position by sodium atoms. (**c**) Average sodium number during the simulation as a function of the tube axis (red for (10,10) SWCNT, black for (13,13) SWCNT functionalized by 5 –COO^−^ functions, respectively). (**d**) Autocorrelation function calculated as a function of the correlation time at the direct entrance of each SWCNT, where the density of Na^+^ ions is the largest one.

**Table 1 nanomaterials-14-00117-t001:** Force field (stretching and bending moduli and equilibrium angles) and Lennard–Jones parameters for the –OH and –COO^−^ functions.

Force Field parameters for functions added to CNT			
Bonds	*K_b_* (kcal/mol/Å^2^)	Angles	*K_a_* (kcal/mol/A^2^)
C–C (r_0_ = 1.415 Å)	600	C–C–C (θ_0_ = 120°)	350
C–O (r_0_ = 1.395 Å)	300	C–C–O (θ_0_ = 120°)	45
C–H (r_0_ = 1.415 Å)	600	H–O–C (θ_0_ = 119.47°)	35
H–O (r_0_ = 0.948 Å)	605	H–C–C (θ_0_ = 118.85°)	600
**Lennard Jones parameters:**			
**Atom**	**σ (Å) (R** ** _min_ ** **/2)**	**ɛ (kcal/mol)**	
C	1.9474	−0.0660	
H	1.459	−0.0150	
O	1.721	−0.0750	

**Table 2 nanomaterials-14-00117-t002:** Simulated conductances $G_{exp}$ (in nS) for different charged SWCNT with ends unfunctionalized, functionalized with –OH at low functionalization rate or high functionalization rate and functionalized with –COO^−^ at low functionalization rate or high functionalization rate.

	No Function	–OH Functions		–COO^−^ Functions	
		Low	High	Low	High
(10,10)	0.70 ± 0.07	0.16 ± 0.03	0.04 ± 0.02	0.40 ± 0.04	0.16 ± 0.03
(11,11)	0.81 ± 0.04	0.53 ± 0.09	0.44 ± 0.05		
(13,13)	1.70 ± 0.02	1.18 ± 0.02	0.80 ± 0.01	0.72 ± 0.02	0.59 ± 0.01

**Table 3 nanomaterials-14-00117-t003:** Theoretical access conductances (in nS) deduced from Equation (4) and the experimental conductances given in Table 2.

	No Function	–OH Functions		–COO^−^ Functions	
		Low	High	Low	High
(10,10)	1.9 ± 0.5	0.23	0.041	0.62	0.19
(11,11)	1.8 ± 0.2	0.82	0.62		
(13,13)	6.2 ± 0.3	2.38	1.2	1.0	0.79

**Table 4 nanomaterials-14-00117-t004:** Theoretical conductances (in nS) for different charged SWCNT with ends unfunctionalized, functionalized with –OH at low functionalization rate or high functionalization rate or for uncharged SWCNT.

	(19,0)	(23,0)
**Charged unfunct.**	0.81 ± 0.03	1.56 ± 0.04
**Charged funct. (10 –OH)**	0.48 ± 0.05	1.21 ± 0.06
**Charged funct. (16 –OH)**	0.25 ± 0.02	0.99 ± 0.03
**Uncharged unfunct.**	0.10 ± 0.02	0.46 ± 0.01

## Data Availability

Data are contained within the article.

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
