# Peer review of "Impact of Single-Walled Carbon Nanotube Functionalization on Ion and Water Molecule Transport at the Nanoscale"

_nanomaterials, 2024, doi:10.3390/nano14010117_

Round 1
Reviewer 1 Report
Comments and Suggestions for Authors
I've made the following thorough comments:
1-Why did you choose two functional groups (OH and COO) in this study? And why did you not use another functional group on CNT?
2-There are some grammar errors. It is recommended to check and correct English grammar carefully.
3-Why is nanofluidics considered promising, and in which domains does it show potential applications?
4-What challenges are associated with understanding and controlling the behavior of solvents in nanometric channels?
5. Figures 2 and 3 should have better quality. Figure 3 is unclear, making it difficult for the reader to understand.
6- How is water and ion transport in carbon nanotubes investigated in this study?
7- What is the role of the carbon nanotube in the simulation setup, and how does it mimic experimental nanopore devices?
8- How is the external electric field applied in the simulation, and what does it aim to achieve?
9-What parameters related to carbon nanotubes are investigated in the study, and how do they affect water and ion transport?
10-What is the significance of analyzing the evolution of electrolyte conductivity in the context of this study?
11-How does the functionalization of carbon nanotube entrances with OH or COO- -groups impact the study's outcomes?
12. The quality of Tables 2 and 3 should be higher. It is advised to enhance the tables' quality before publication.
Best Regards
Comments on the Quality of English Language
Dear Editor of Nanomaterials journal:
I trust this message reaches you in good health. Recently, I had the chance to assess the manuscript titled " Impact of the single walled carbon nanotube functionalization on the ion and water molecules transport at the nanoscale " which was submitted to the Nanomaterials journal. I commend the valuable insights presented in the manuscript; however, I wish to bring attention to a concern regarding the clarity of the English language. Although the content and scientific contributions are noteworthy, I recommend that the authors focus on enhancing the clarity and coherence of their English expressions. This effort would not only improve the manuscript's readability but also facilitate a more effective communication of their research findings. I propose that the authors consider enlisting the assistance of a professional language editor or proofreader to refine the language quality. Such a step could significantly enhance the overall presentation of their research. Thank you for considering this feedback, and I eagerly anticipate the opportunity to review an improved version of the manuscript.
Best Regards,
Dr. Masoud Khaleghiabbasabadi
A faculty member of the Department of Environmental Chemistry of the Institute for Nanomaterials, advanced technologies, and innovations, technical university of Liberec (CXI)
masoud.khaleghiabbasabadi@tul.cz
Author Response
Dear editor
We would like to thank the two reviewers for their careful review of our article. We have taken into consideration all their remarks in this new version. The answers to each major revision issues are detailed below.
(1) Why did you choose two functional groups (OH and COO) in this study? And why did you not
use another functional group on CNT?
Author reply: The impact of many different functional groups could be studied. For the first investigation on that subject, we have chosen only two which seems to us having interesting features: the carboxylate (-COO-) which is stable at pH=7 and alcohol functions (-OH). This allows us also to analyze the influence of non-charged chemical functions compared to highly charged ones on the ionic conductance behavior.
We added a small paragraph in the model section to explain it line 97.
(2) There are some grammar errors. It is recommended to check and correct English grammar
carefully.
Author reply: done. A native English speaker has read and corrected the paper
(3) Why is nanofluidics considered promising, and in which domains does it show potential
applications?
Author reply: Carbon nanotubes (CNTs) with nanoscale diameter present a strong hydrophobic surface providing a strong confinement and unusual water behavior. This leads to many unexpected properties, such as flow enhancement, strong ion exclusion, ultrafast proton transport and phase transition. These properties are applied in different domains allowing to develop applications in nanofiltration for desalination system or in nanosensor as for single molecule or DNA detection.
(4) What challenges are associated with understanding and controlling the behavior of solvents in
nanometric channels?
Author reply: Fluid and ion transport through CNT is the subject of intense research due to the strong hydrophobic properties of the carbon wall. This leads to unexpected but remarkable behaviors in nanofluidic systems that are still now not fully understood. Experimentally, the size of the nanodevice, based on individual nanochannel leads to high variability of the experimental conditions and thus high changes in the observed results. The goals of the simulations at the atomistic scale and of the theoretical developments are precisely to determine and understand the role of each parameter on the nanofluidic device properties.
(5) Figures 2 and 3 should have better quality. Figure 3 is unclear, making it difficult for the reader
to understand
Author reply: Thanks to the referee remark, all Figures have been redrawn for clarity.
6) How is water and ion transport in carbon nanotubes investigated in this study?
Author reply: Water and ion transport are studied through all atom molecular dynamic simulations under applied electric field. This allows us to calculate the ionic flow through the hydrophobic CNT nanochannel and compare these simulation results to theoretical investigations.
7- What is the role of the carbon nanotube in the simulation setup, and how does it mimic
experimental nanopore devices?
Author reply: The CNT is precisely the heart of the nanodevice. It is used as the ultimate nanopore channel in several nanofluidic systems. This makes the experiments so difficult to perform and to interpret. This is precisely why simulations and theory are necessary to help experiments in their developments.
8- How is the external electric field applied in the simulation, and what does it aim to achieve?
Author reply: Technically, the electric field is applied through the electrolyte, along the CNT axis, such that each ions of the electrolyte undergoes a constant electric force. Ion species therefore move along this direction which creates an electric current measured between the two electrodes responsible for this electric field.
9-What parameters related to carbon nanotubes are investigated in the study, and how do they affect water and ion transport?
Author reply: In this paper the role of the chemical functionalization of the dangling bonds and the CNT diameter have been investigated in order to understand their influence on the ionic conductance through the CNT. We show, as discussed in the paper, and summarized in the small conclusion added in the revised manuscript, that the charge of the chemical functions, but also its steric volume change the access conductance and make the ionic conductance strongly modified compared to nude extremities.
10-What is the significance of analyzing the evolution of electrolyte conductivity in the context of
this study?
Author reply: The precise goal of our study is to know how the ionic flow through the hydrophobic CNT channel is modified depending on external chemical functionalization of the channel. It aims at understanding the variation of the ionic conductance through the application of a voltage difference. The electrolyte conductivity is one of the major transport coefficient for electrolytes and is relatively simple to measure experimentally.
11-How does the functionalization of carbon nanotube entrances with OH or COO- -groups
impact the study's outcomes?
Author reply: We show, as discussed in the paper, and summarized in the small conclusion added in the revised manuscript, that the charge of the chemical functions, but also its steric volume, change the access conductance and make the ionic conductance strongly modified compared to nude extremities.
- The quality of Tables 2 and 3 should be higher. It is advised to enhance the tables' quality
before publication.
Author reply: We thank the Referee for his/her advice. All Tables have been redesigned for clarity.
We hope now that our new version will satisfying to be accepted for publication.
Reviewer 2 Report
Comments and Suggestions for Authors
In this manuscript the authors investigate water and ion transport in single-walled carbon nanotubes (SWCNTs) using classical force field molecular dynamics (MD) simulations. A single nanopore device similar to experimental ones reported in the literature is mimicked by combining one SWCNT with two perforated graphene sheets. Conductivity of this system is calculated and analyzed with respect to SWCNT geometric parameters (radius and chirality), functionalization of the CNT entrances with OH or COO groups depending on the concentrations of group functions. The authors conclude that their simulations could be verified by experiments, thus explaining why nanofluidic experiments lead to significant conductivity differences that depend on experimental conditions. The paper is potentially useful for the nanofluidics community, but it exhibits several weaknesses that make it unsuitable for publication in its current form.
Major issues:
(1) Regarding the total conductance of the CNT system: does it include electronic conductance of the tube itself, in addition to the conducting ions through the CNT? How is the current simulated, which transport model(s) is used? How do you motivate using classical force field molecular dynamics (MD) simulations for such nanoscale structures with strong quantum effects?
(2) How are contacts treated in the simulations? How realistic is this treatment?
(3) Related question: if electronic conductance of CNT is not included, how do you explain increase of current in CNTs with smaller radius reported in Fig.3?
(4) Does access resistance have anything to do with contact resistance coming from a limited number of conducting modes in CNTs and other 1D carbon nanostructures? See e.g. DOIs: 10.1039/C6NR01012A and 10.3390/ma14133670
(5) The paper is missing a comprehensive but succinct conclusion section to summarize a great number of results presented in the paper
Minor issues:
(1) page 1: section 0 from the template must be removed before submission
(2) page 4: typo in line175 (a question mark appears for no reason)
(3) poor figure and table quality, these must be improved: improve figure quality for Fig. 2; Fig. 3 is not readable at all; Fig. 5 and Fig 7 are barely readable
(4) not all equations are numbered
Comments on the Quality of English LanguageGenerally understandable, minor editing needed.
Author Response
Dear editor
We would like to thank the two reviewers for their careful review of our article. We have taken into consideration all their remarks in this new version. The answers to each major revision issues are detailed below.
Response to reviewer 2:
(1) Regarding the total conductance of the CNT system: does it include electronic conductance of the tube itself, in addition to the conducting ions through the CNT? How is the current simulated, which transport model(s) is used? How do you motivate using classical force field molecular dynamics (MD) simulations for such nanoscale structures with strong quantum effects?.
Author reply: Referee is right, we did not consider the electronic conductance coming from the carbon nanotube wall. We are only interested in the ionic conductance of the electrolyte which is confined in the nanotube. The ionic conductance is investigated owing to an electric field applied through the electrolyte.
Indeed, in our simulations, we did not consider quantum effects because of the huge size of the system and because we are only interested in the ionic conductance. If we want to approach the experimental system size, with the electrolyte reservoirs treated in full atom representation, the only possible simulations are classical ones.
The fruitful comparison of our results with experimental and theoretical ones shows that our classical approach is not so simplistic. If we compared with a recent review (https://doi.org/10.1021/acs.jpcc.1c08202), written by our consortium and summarizing all the experimental data in such field of research, we can observe that our simulations results are well matching the table given in the supplementary data of this article or in Fig 1 of this article.
(2) How are contacts treated in the simulations? How realistic is this treatment?
Author reply: In these simulations and calculations, we are only interested in the ionic transport inside the nanopore due to an external voltage drop only applied to the electrolyte, which is experimentally done thanks to electrodes in the two reservoirs (of course not simulated here).
(3) Related question: if electronic conductance of CNT is not included, how do you explain increase of current in CNTs with smaller radius reported in Fig.3?
Author reply: The behaviour is ohmic (i.e. the current increases linearly with the voltage drop) with an ionic conductance which is independent on the voltage drop but which depends not only on the nanotube radius but also on the electrical charge on the nanotube and on the chemical function attached at its extremities. Note that we show in Fig. 3 that the conductance decreases when the pore radius decreases.
(4) Does access resistance have anything to do with contact resistance coming from a limited number of conducting modes in CNTs and other 1D carbon nanostructures? See e.g. DOIs: 10.1039/C6NR01012A and 10.3390/ma14133670
Author reply: The access resistance in this paper corresponds to the decreases of the surface normal to the current when ions are going from the reservoir to the nanopore, and as nothing to do with contact resistance.
(5) The paper is missing a comprehensive but succinct conclusion section to summarize a great number of results presented in the paper
Author reply: We thank the referee for this good remark. We have added a small paragraph at the end of the paper to summarize the main conclusions of this work.
We hope now that our new version will satisfying to be accepted for publication.
Round 2
Reviewer 1 Report
Comments and Suggestions for Authors
Dear Sir/Madam
I hope this message finds you in good health. I am writing to express my positive evaluation of the revised manuscript titled "Impact of the single walled carbon nanotube functionalization on the ion and water molecules transport at the nanoscale" which I had the privilege of reviewing.
The author has diligently addressed all the feedback and comments provided during the review process. The revisions made have significantly strengthened the manuscript, and in my assessment, the paper is now in excellent form for publication.
The clarity of presentation, adherence to scholarly standards, and overall quality of the content have been notably enhanced. I believe that this paper will provide a valuable contribution to Nanomaterials, offering meaningful insights to the readership.
I recommend accepting the revised manuscript for publication. I appreciate the author's commitment to refining the work and the journal's dedication to maintaining high standards of academic excellence.
Thank you for considering my feedback, and I eagerly anticipate seeing this insightful contribution published in Nanomaterials.
Sincerely,
Reviewer 2 Report
Comments and Suggestions for Authors
All comments have been addressed in more or less detail. English is improved. Manuscript looks OK for publication.